# Tapping into Plant–Microbiome Interactions through the Lens of Multi-Omics Techniques

**DOI:** 10.3390/cells11203254

**Published:** 2022-10-17

**Authors:** Ajay Kumar Mishra, Naganeeswaran Sudalaimuthuasari, Khaled M. Hazzouri, Esam Eldin Saeed, Iltaf Shah, Khaled M. A. Amiri

**Affiliations:** 1Khalifa Centre for Genetic Engineering and Biotechnology, United Arab Emirates University, Al Ain P.O. Box 15551, United Arab Emirates; 2Department of Chemistry (Biochemistry), College of Science, United Arab Emirates University, Al Ain P.O. Box 15551, United Arab Emirates; 3Department of Biology, College of Science, United Arab Emirates University, Al Ain P.O. Box 15551, United Arab Emirates

**Keywords:** plant–microbe interactions, rhizosphere, root exudates, primary metabolites, secondary metabolites, allelopathy, plant microbiome, metagenomics, metatranscriptomics, metabolomics, sustainable agriculture, ecology

## Abstract

This review highlights the pivotal role of root exudates in the rhizosphere, especially the interactions between plants and microbes and between plants and plants. Root exudates determine soil nutrient mobilization, plant nutritional status, and the communication of plant roots with microbes. Root exudates contain diverse specialized signaling metabolites (primary and secondary). The spatial behavior of these metabolites around the root zone strongly influences rhizosphere microorganisms through an intimate compatible interaction, thereby regulating complex biological and ecological mechanisms. In this context, we reviewed the current understanding of the biological phenomenon of allelopathy, which is mediated by phytotoxic compounds (called allelochemicals) released by plants into the soil that affect the growth, survival, development, ecological infestation, and intensification of other plant species and microbes in natural communities or agricultural systems. Advances in next-generation sequencing (NGS), such as metagenomics and metatranscriptomics, have opened the possibility of better understanding the effects of secreted metabolites on the composition and activity of root-associated microbial communities. Nevertheless, understanding the role of secretory metabolites in microbiome manipulation can assist in designing next-generation microbial inoculants for targeted disease mitigation and improved plant growth using the synthetic microbial communities (SynComs) tool. Besides a discussion on different approaches, we highlighted the advantages of conjugation of metabolomic approaches with genetic design (metabolite-based genome-wide association studies) in dissecting metabolome diversity and understanding the genetic components of metabolite accumulation. Recent advances in the field of metabolomics have expedited comprehensive and rapid profiling and discovery of novel bioactive compounds in root exudates. In this context, we discussed the expanding array of metabolomics platforms for metabolome profiling and their integration with multivariate data analysis, which is crucial to explore the biosynthesis pathway, as well as the regulation of associated pathways at the gene, transcript, and protein levels, and finally their role in determining and shaping the rhizomicrobiome.

## 1. Introduction

The burgeoning world population is predicted to reach 8.5 billion by 2025, roughly representing a one-percent increase [1]. This alarming condition underscores the need to rapidly and continuously increase plant–based food, feed, and fiber production. Food production can be augmented by expanding the cultivation area and improving the yield per unit area sown by reducing pre- and post-harvest losses and implementing effective weed control strategies. Growers resort to herbicides as one method of minimizing weed infestation and protecting their crops. Many factors contribute to an effective herbicide program, including timing of application, use of total labeled rates, appropriate adjuvants, and efficient spray application [2]. Improper or continuous applications of similar herbicides to the same agricultural land over several years and the number of herbicides used for weed control are some factors that can lead to herbicide resistance [2]. In this context, the potential of biologically active plant root exudates in the interaction (intra- or inter-species) and recruitment of microbial communities could be exploited as environmentally friendly strategies for crop growth and protection against weeds, insects, and pathogens.

Plants are colonized by various microorganisms, both below and above ground simultaneously, with mutualistic benefits that not only assist each other in obtaining resources or services, but also serve as a vital factor in shaping the dynamics of interacting populations [3]. In plants, diverse microbial communities have been identified in the phyllosphere (epiphytes) [4], the rhizosphere [5], and the endosphere (endophytes) [6]. Among them, the rhizosphere microbial community is a complex with a dynamic spatiotemporal structure (a phenomenon known as the “rhizosphere effect”) that rapidly adapts in response to fluctuations in metabolites exuded by plant roots and to a variety of environmental factors, including soil composition, temperature, and vegetation, which in turn generate a considerable influence on nutritional status and growth of plants [7]. The majority of root exudates include low-molecular weight primary metabolites (sugars, amino acids, and organic acids) [8] and secondary metabolites (alkaloids, cyanogenic glycosides, flavonoids, and terpenoids) [9], which have been shown to play pivotal roles in various biological processes such as mobilization and acquisition of nutrients [10], qualitative and quantitative composition of the microbiome in their immediate vicinity [11], inhibition of the growth of competing plant species, and governing plant survival under abiotic and biotic stresses [12]. Root exudation is a major source of organic carbon in the soil, accounting for up to 30–60% of total photosynthetic production [13]. Exudates can therefore serve as a significant energy source for soil microorganisms, which are frequently carbon-limited. This results in a process known as “soil priming”, in which the microbial community becomes more active and releases nutrients crucial for plant health [14], similar to probiotic bacteria in the digestive system of vertebrates.

Furthermore, there is substantial evidence that root exudates can improve soil stability and resistance to mechanical and hydraulic stresses. The previous study demonstrated that incorporating maize root and chia seed exudates increased aggregate stability, whereas barley root exudates decreased soil tensile strength and aggregate stability [15]. In order to facilitate root extension and penetration into the soil, the root exudates also reduce the fractional contact between root surfaces and soil particles [16].

The quantity, quality, and composition of root exudates depend on various endogenous (species, varieties, developmental stages, and functional characteristics of plants) and exogenous biotic (rhizosphere microbial community, herbivores, and neighboring plants) and abiotic factors (temperature, light and water, soil texture, soil pH, soil organic matter, moisture, and nutrient supply) [17]. In rice, for example, exudation rates are lowest at the seedling stage, increase until flowering, and then decrease at maturity [18]. A significant decrease in the composition of root exudates was observed in two different *Lupin* species (*L. albus* cv. Multolupa and *L. luteus* cv Tremosilla) from flowering to fruiting stages [19]. Similarly, in wheat and sorghum, root exudation decreases with plant age and increases in response to soil stress caused by drought, compaction, and nutrient deficiency [18]. A recent study showed the impact of different exogenous factors on the composition of polar and semi-polar metabolites of root exudate in perennial grass and forb species with higher chemical richness [20].

The quantitative and qualitative composition of exudates depends on root surface morphology (periderm thickness, suberization, root hairs density and location, mycorrhizal hyphae, etc.), root system architecture (such as root length, branching, number and length of lateral roots) and the actively growing root system [21]. Exudation compositions depend on the root zone. The root cap, root hair cells, and the zone behind the root tip all actively participate in the exudation process, followed by stellar and cortical cells [22]. All these variables interact to influence root exudation and microbial community composition. 

This review article aims to provide, for the first time, a sweeping view of roots–rhizosphere interactions, as well as plant–plant interactions governed by multipartite chemical communications involving the exudation of specialized primary and secondary metabolites, including allelochemicals. Moreover, we attempt to provide insights into various omics approaches such as genomics, metagenomics, and microbial genome-wide association studies, including metabolomics and multivariate data analysis in the context of plant signaling metabolite discovery and identification of causal relations of plants with microbial communities. In this concise review, we have attempted to showcase the most important past and recent discoveries in the field of plant–microbe and plant–plant interaction at the interface of secreted metabolites and cutting-edge technologies to better understanding the mechanism, which is indispensable for researchers working in plant–microbe interactions.

## 2. Primary Metabolite: Chemical Currency with Multipartite Function

Plants require essential mineral nutrients such as phosphorus (P), nitrogen (N), potassium (K), and sulfur (S) for the biosynthesis of proteins, enzymes, vitamins, chlorophyll, nucleotides, and other metabolites. Plants absorb up to 90% of organically bound mineral nutrients from soil with the help of microorganisms (e.g., symbionts) and transfer 30–60% of their photosynthetic products (about 5 billion tons of carbon per year) to the rhizosphere via root exudation of primary metabolites [13]. The allocation of the majority of photosynthesized carbon to root exudates remains ambiguous. According to current ecological theories, photosynthesized carbon exudation serves as a chemical currency and promotes plant productivity by enhancing nutrient acquisition and water-holding capacity, facilitating efficient drainage and aeration, and providing substrates for microbes [23]. In addition to improving soil health, the allocation of fixed carbon to the root exudation process reduces global warming caused by rising CO_2_ levels in the atmosphere [24]. Sugar and organic compound exudation into the rhizosphere can stimulate the decomposition of previously stabilized inherited soil organic matter and fosters nutrient acquisition and soil aggregation [25]. In contrast, exudation of amino acids contributes to the structure of rhizospheric microbial communities, shaping root system architecture, and is critical for nutrient mobility throughout the plant’s vegetative cycle [26].

The mechanism of primary metabolite exudation from roots is still equivocal. According to a recently proposed concept, primary metabolites are released into the soil through a source–sink-driven diffusion process mediated by the modulation of nutrient concentration gradients between plants and the soil environment or via efflux carriers, rather than as an uncontrolled passive leakage [10]. The majority of the primary metabolite flux occurs near the root tip, where a cluster of undifferentiated cells promotes metabolite diffusion into the soil depending on the source level (phloem loading and vertical transport) or at the sink level (phloem unloading, metabolism, exudation, and microbial consumption) [10]. Moreover, transcriptional regulation and post-translational modification of specific efflux transporters and channels facilitate the fine-tuned exudation of sugars, amino acids, and organic acids [27]. Over the past decades, several candidate efflux membrane transporters involved in the exudation of primary metabolites such as organic acids MATE/citrate transporters [28], sugars (SWEET sugar transporter family [29]), and amino acids (Glutamine Dumper1 [30], Cationic Amino Acid Transporter family, Lysine and Histidine Transporter 1, Proline Transporter 2, Neutral Amino acid Transport system 2) [31] have been discovered and characterized. The phloem sap contains a variety of other primary metabolites, and our current understanding of their role in the plant–microbe interaction and the role of the associated efflux transporter system is still limited. Primary metabolites, including hormone trafficking, depend on plasmodesmata and membrane transport proteins. In this realm, synchronous capture of eukaryotic RNAs with polyA tails in conjunction with transcriptomic profiling at the single-cell level could provide insight into their role and allocation. Furthermore, genetically encoded fluorescent compartment-targeted biosensors will allow for the dissection of spatial dynamics of primary metabolites, transporter activities, and biophysical parameters within cells and across membranes [32].

## 3. Secondary Metabolite: Microbial and Plant Community Modulator Trait

Plant secondary metabolites (PSMs) are a diverse group of compounds derived either from primary metabolites or intermediates of primary metabolite biosynthetic pathways through the sequential action of specialized enzymes regulated by phytohormones and signaling molecules. Based on chemical structure and biosynthetic pathway, PSMs are categorized into three major classes: (i) terpenoids (terpenes, sterols, glycosides, saponins, carotenoids, and steroid), (ii) polyphenols (flavonoids, phenolic acids, tannins, stilbenes, lignans, and coumarins. etc.), and (iii) nitrogen-containing compounds (amines, alkaloids, cyanogenic glycosides, and glucosinolates) [33]. Unlike primary metabolites, PSMs do not directly contribute to the primary functions of growth and development. Nevertheless, they play an essential role in plant response to various abiotic and biotic stresses, underground interactions, shaping microbial communities, communication (microbe–microbe, plant–microbe, and plant–plant), and adaptation [34]. 

Previous research on legume metabolites and their mediated interactions provided the best model for the interaction between PSM and microbes. Several studies have documented the significant abundance (>70%) of rhizobia in the root microbiome of legumes (such as beans, peas, peanuts, chickpeas, lentils, lupins, and soybeans) than in bulk soil [35,36]. The function of certain flavonoids (polyphenolic PSM) in establishing mutualistic relationships with nitrogen-fixing *Rhizobium* spp., actinorhizal, and arbuscular mycorrhizal fungi has been extensively studied in the legume family [37,38,39]. Depending on their structure, flavonoids can stimulate or inhibit rhizobial nod gene expression, control nodule development and differentiation, and cause significant alteration in microbiome assembly and composition in the rhizosphere and rhizoplane by interfering with co-occurrence interactions and substantially depleting root microbes [39]. In addition, the silencing experiment designed to silence gene encoding chalcone synthase in *Medicago truncatula* root showed that flavonoids are also involved in the local accumulation of auxin at the nodule formation site for initiation of nodule primordia [40]. Legume roots secrete flavonoids/isoflavones as signaling compounds to entice nitrogen-fixing bacteria such as *Rhizobium*, *Bradyrhizobium*, *Neorhizobium*, *Allorhizobium*, *Azorhizobium*, *Mesorhizobium*, *Pararhizobium*, and *Trinickia* to entrench host plants through root hairs and subsequently multiply and stimulate the formation of root nodules [35,41]. In this process, the diversity and concentration of flavonoids in the root exudates of legume species form the fine-tuned molecular signal and thus act as a selection factor for determining symbiosis specificity (compatible or incompatible) in plants. For instance, the isoflavonoid medicarpin secreted by *Medicago* and *Trifolium* species exhibits an antagonistic effect on the growth of certain bacterial strains such as *Bradyrhizobium japonicum* and *Mesorhizobium loti* and a positive chemotaxis on *Rhizobium melioti* [42]. Similarly, the symbiotic compatibility or non-compatibility of *Frankia* spp. depends on the flavonoid composition of the fruit exudates of the actinorhizal plant *Myrica gale* [43].

Terpenoids represent the largest group of specialized PSM. Besides their essential functions in ecological adaptation, defense, growth, and development, they bestow the chemical language to plants to communicate with bacterial and fungal communities [44]. The activities of triterpenoids in establishing rhizosphere communities were demonstrated in *A. thaliana*, where mutant lines defective in triterpene and sesterterpene biosynthesis formed distinct root microbiome communities compared to the wild type [45]. The proportional relationship of alkaloids influencing the composition and diversity of endophytic bacterial communities has been shown in the root, stem, leaf, and fruit of *Macleaya cordata* [46]. During host colonization, filamentous fungi and oomycetes release a diverse array of glycoside hydrolases (GHs) as virulence factors (effectors) onto their cell surfaces and the surrounding extracellular environment, which plants utilize during host microbial colonization [47]. In addition to the aforementioned soluble secondary metabolites, plants also release diverse volatile organic compounds (such as phenylpropanoids, terpenoids, benzenoids, and β-caryophyllenes) that diffuse through the air- and water-filled pores in soil interactions, which further interconnect the plant rhizobiomes through microbial metapopulation networks [48]. Volatile organic compounds (VOC) predominantly influence soil microbiomes, and contrasting differences in their richness in soil may be correlated with the magnitude of the signal produced by the VOC [49]. Although volatile chemicals can promote intra- and interspecific interactions and alter microbial and plant fitness, it is still unclear how microbial communities influence VOC signaling as it travels through the soils.

Certain classes of secondary metabolites are classified as allelochemicals. Based on the biochemical pathways, the chemical structure and properties of allelochemicals can be classified into 14 categories: (i) water-soluble organic acids (straight-chain saturated primary alcohols, aliphatic aldehydes, and ketones), (ii) α,β-unsaturated aromatic lactones, (iii) long-chain fatty acids and polyacetylenes, (iv) quinines (benzoquinone, anthraquinone, and complex quinones), (v) simple phenols, benzoic acid, and its derivatives, (vi) cinnamic acid and its derivatives, (vii) coumarin, (viii) flavonoids, (ix) tannins, (x) steroids and terpenoids (sesquiterpene lactones, diterpenes and triterpenoids), (xi) amino acids and peptides, (xii) alkaloids and cyanohydrins, (xiii) sulfides and glucosinolates, and (xiv) purines and nucleosides [50]. A growing body of research has implicated the role of plant-derived allelochemicals in various interactions, including microbe–microbe, plant–microbe and plant–plant [51,52].

Differences in microbial density have been observed in different parts of the rhizosphere and correlate with the amount and composition of allelochemicals released by specialized root cells [17]. The composition and quantity of compounds synthesized in the plant and exuded from the root are under the genetic control of the plant. Therefore, changes in the composition of root exudates within and among plant species significantly modulate density, composition, and biological activity of microbial communities, resulting in plant–dependent selection of microbial communities [53,54]. Although the mechanisms by which these compounds affect the microbial community have remained equivocal. A growing body of research has shown that the amount and form of carbon (soluble sugars), amino acids, or secondary metabolites that the plant releases into the soil, including allelochemicals, create a niche for specific rhizospheric microbes, which in turn may have a feedback effect on the colonization of conspecific and heterospecific microbes growing in the same soil [55]. Plants exert maximum selective pressure on the microbial population near the root surface or in the root interior. Thus, allelochemical-mediated selective pressure shapes community composition and microbial diversity in the rhizosphere in a plant-dependent manner [55].

The specificity of the rhizosphere microbiome of *A. thaliana* has been linked to differences in exudate profiles [56]. The roots of wheat (*Triticum aestivum* and *T. durum*), rapeseed (*Brassica napus*), maize (*Zea mays*), and barrel clover (*Medicago truncatula*) harbor different bacterial communities owing to differences in the chemical composition of their respective root exudates [57]. Similarly, significant variations in the bacterial community (in terms of richness, relative abundance, and diversity) were found in the rhizosphere of 27 inbred maize lines, which correlated with differences in root exudate profiles [58]. Plant allelochemicals not only influence the rhizospheric microbial community but can also drive and shape the selection of soil microbes. For instance, the application of *p*-coumaric acid, a well-known allelochemical, to soil-grown cucumber seedlings enhanced the relative rhizosphere bacterial (*Betaproteobacteria*, *Firmicutes*, *Gammaproteobacteria*) and fungal (*Sordariomycete* and *Zygomycota)* community abundances and decreased the relative abundances of bacterial taxa such as Deltaproteobacteria, Planctomycetes, Verrucomicrobia and fungal taxon Pezizomycete in the cucumber rhizosphere. It increased the population density of a soil-borne pathogen (*Fusarium oxysporum* f.sp. *cucumerinum*) of cucumber [59]. Another study showed that allelochemicals released by plants of the Brassicaceae, Caryophyllaceae Chenopodiaceae, and Cyperaceae families suppressed fungal pathogen spores germination and disrupted symbiotic associations between mycorrhizal fungi with nearby growing host plants [60].

The soil microbiota plays an essential role in determining the structure and dynamics of plant communities and can profoundly influence ecosystem invasion by exotic plant species. Previous research on invasive plants has revealed a wide variety of plant–soil interactions that might enhance invasiveness in a new range [61,62,63]. Two distinct mechanisms by which plant–soil community interactions influence native plant dominance and exotic plant invasiveness have been proposed [64]. First, exotic plants can manipulate the local soil biota by increasing pathogen levels or disrupting root symbiont communities. For example, chemical compounds (allyl isothiocyanate) released by the invasive species garlic mustard (*Alliaria petiolate*) disrupt the association of ecto- and arbuscular mycorrhizal fungi with roots of native plants [65]. Root exudates of the tropical invasive weed, *Chromolaena odorata,* disrupt the soil microbiota and promote the accumulation of the pathogenic fungus *Fusarium semitectum* in its rhizosphere, negatively affecting the growth of neighboring native plant species [66]. In an alternative scenario, native soil communities might be unable to detoxify allelochemicals of exotic plant species or might render them even more toxic through microbial-mediated conversion (Figure 1).

Efflux of secondary metabolites from the cell membrane, including allelochemicals, is facilitated by passive transport (diffusion, ion channels, and exocytosis via secretory vesicles) and an active transport process involving the utilization of specific membrane-bound transport proteins [21].

The last few years have seen unprecedented efforts to understand the role of metabolites in establishing and modulating the structure of the rhizosphere and root (endophyte) microbial community and to identify the members of the soil microbial communities coupled with their relative abundance fluctuation in response by employing several omics approaches such as genomics, metagenomics transcriptomics, proteomics, and metabolomics. The advent of next-generation sequencing (NGS) technologies has revolutionized and accelerated robust high-throughput data generation of genomics, transcriptomics proteomes, and metabolomes, ushering in a new era of big data that pave the way for characterizing and exploring the impact of secreted metabolites on the biology and ecology of plants, plant-associated microorganisms, and their interactions [67]. In the following section, we outlined the recent research on the implementation of omics resources and the integration of multi-omics data in inferring the detailed picture of plant–microbial interactions, as well as the identification and characterization of compounds exudated from the root into the rhizosphere and their role in the interaction with rhizosphere-associated microbes.

## 4. Metagenomics and Metatranscriptomics: A Paradigm Shift in Microbiomics

The last two decades have arguably witnessed unprecedented progress in understanding the role of root morphology and root exudation in shaping rhizobiomes. The first initiative in this field was the identification of microbes in root exudates using traditional culturing methods [68]. Since the introduction of first-generation sequencers, amplicon-based strategies targeting variable regions of genomes (such as 16S, ITS, or 18S) have been widely used to describe the composition of bacterial, archaeal, fungal, and micro-eukaryotic communities in the rhizosphere [69]. However, cataloging the vast diversity of the rhizosphere owing to unculturable microbes, their host and habitat preference has challenged this method. The advanced power of DNA sequencing technology (second-generation sequencers) circumvents these obstacles by cost-effective DNA sequencing isolated directly from the soil, rhizosphere, and root samples [70]. Studies leveraging next-generation sequencing (NGS) technologies for amplicon sequencing of the 16S rRNA gene, ITS/18s rRNA, have led to the field of metagenome analysis [71,72]. In recent years, community structure and diversity of soil microbes have been the focus of soil ecology research. Sixty-four sequences of the soil microbiome of rice have been characterized by metagenomics [73]. Nevertheless, the metagenome-based approach provides new insights into the variation of fungal populations, bacterial diversity, and their richness in both the bulk soil and rhizosphere of soybean [74], the effects of continuous sugar beet cultivation on its endophytes [75] and the assessment of rhizosphere diversity and endophytic fungi in *Atractylodes macrocephala* [76].

This PCR amplicon-based approach follows the steps such as NGS data QC and trimming, read merging (for PE reads), OTU/ASV table generation based on the reference database Greengene [77], Silva [78], and Genbank [79], community profiling (alpha and beta diversity analysis), statistical analysis (correlation analysis, clustering and heatmap) and functional analysis using PICRUSt [80] and Tax4Fun [81] (Figure 2). Long-read sequencing technologies (PacBio and Oxford Nanopore Technology) have recently been used to profile microbial communities in environmental samples at species-level resolution by sequencing the entire marker gene (16s, 18s rRNA and ITS) amplicon [82,83]. This amplicon-based method is widely used for exploring microbial profiling from samples. In a recent study, 16s rRNA-based microbiota analyses of rice rhizosphere and seed identified the vertical transformation of microbes from parental seed to offspring [84].

An essential step in the identification of plant–microbe interactions is the sequencing of the genome of microbes associated with or surrounding the plant [85]. The whole genome of the organism provides information on the entire set of genes present in the organism, key enzymes, and associated metabolic pathways, thus improving our understanding of the biogeochemical cycle of the organism [86]. In this context, metagenome assembly facilitates not only the assembly of individual DNA sequences into genes or organisms to obtain complete genetic information, but also the exploration of hidden resources such as new genes, biomolecules, valuable products, and complex functions of microbial communities and the interactions between these microbes in the rhizosphere [87]. The metagenome assembly process generally includes the following steps: NGS data quality control, genome assembly (de novo or overlap-based), assembly quality control, genome binning, gene prediction, gene annotation, genome profiling, functional genome prediction, and variation analysis (Figure 2). Long-read sequencers (PacBio and ONT) are widely used to generate high-quality metagenome assembled genomes (MAGs) from metagenome samples [88]. MAGs compiled from the samples provide detailed insight into the functional behavior of the microbes found in the samples, including high-resolution microbial profiling of the samples under study [88]. Using this technique, researchers discovered differences in microbial composition (bacteria, fungi, and archaea) between the healthy and diseased tomato rhizospheres samples [89]. Metatranscriptomics is an RNA-based method that provides valuable information about the entire gene expression profile of complex microbial communities and the impact of environmental conditions on the diversity and expression profile within such a community [90]. This technique has been deployed to investigate changes in microbiomes in the soil and rhizospheres of wheat, oat, pea, and an oat mutant [91] as well as symbiotic activities of ectomycorrhizal fungi [92]. Metatranscriptome data analysis mainly involves quality control, transcriptome assembly, taxonomic profiling, functional annotation of transcript, pathway analysis, and differential gene expression analysis (Figure 2). Shotgun data are used for metatranscriptome studies, but in recent times, long-read sequencers have been used for sequencing the metatranscriptomics samples [93].

## 5. Synthetic Microbial Communities (SynComs) Tool: An Exciting Frontier in Rhizosphere Research

In recent years, plant microbiome research has rapidly advanced in understanding the complexity and dynamics of plant–microbe and microbe–microbe interactions in the rhizosphere. The availability of such information is being used in the formulation of microbial inoculants as an effective complementary or alternative tool for agricultural sustainability and productivity [94]. However, the practice demonstrated inconsistent efficacy in field conditions, which can be attributed to ineffective colonization and interaction of microbial inoculants with the plant, soil microbiome, and soil environment [95]. The lack of mechanistic understanding of the underlying processes of colonization and establishment is a bottleneck to the successful implementation of conventional inoculants in current agricultural practice [95]. An exciting development in this area is the development of versatile and scalable synthetic microbial community (SynCom) systems, where defined and controlled microbial communities are assembled from an extensive collection of microbes to advance our understanding of microbial interactions within diverse natural and artificial microbiomes, including the host [96]. The rationale is to reduce the complexity of the microbial community by tracking and monitoring changes in microbial inoculants in the environment for reproducible implementation in field conditions to promote robustness in terms of colonization and crop production and resiliency against biotic and abiotic stress in agriculture. For instance, the SynCom was constructed from sugarcane-associated and rhizosphere-associated microbes, resulting in increased biomass and improved drought tolerance in maize plants [97] and improved yields and scab tolerance in potatoes [98]. Synthetic microbial systems have recently reawakened interest in plant–microbe interaction research because of their ability to dissect the role of secondary metabolites, signaling molecules, and other components of plant root exudates in the dynamic biological and ecological interactions of microbiomes [99]. The SynCom-based approach has been employed in probing the microbial interactions driven by exometabolites [100], identifying plant genetic factors that determine the relative abundance and composition of the phyllosphere community [101], and studying the role of specialized metabolites on the colonization of microbes in *A. thaliana* and maize rhizospheres [102,103]. Although the SynCom provides a promising platform for studying rhizosphere dynamics and structure, several associated challenges, such as maintaining the long-term stability of SynCom (due to genomic evolution and horizontal gene transfer) and the complexity of dealing with multiple microorganisms and their compatibility with different plant genotypes must be overcome before its large-scale implementation. Integration of gnotobiotic systems can elucidate the functionality of complex consortia and plant phenotype under controlled and reproducible conditions. Furthermore, such use of SynCom can be refined through the use of microfluidics [104], where sensors can be used to detect specific metabolites that respond dynamically to microorganisms.

## 6. Genome-Wide Association Study: Tool for Dissecting the Genetic Basis of Secondary Metabolite Variation

A growing body of research has established that the composition of secondary metabolites in plants is an inherently variable trait. Genetic polymorphisms cause variations in metabolite composition and abundance among cultivars, ecotypes, and species [105,106,107]. For instance, significant variation in the metabolome content of flavonoids, phenolamides, terpenoids, and hydroxycinnamic acid derivatives were found in root exudates of japonica and indica varieties [108]. A large variation of phenylpropanoids, glycosylated metabolites, and plant hormone-derived metabolites was observed in the root exudates of nineteen genetically diverse accessions lines of *A. thaliana* [109]. Similarly, differences in secondary metabolite composition (phenylpropanoids, glycosylated metabolites, plant hormone-derived metabolites) were observed in root exudates of six different ecotypes of *Moringa oleifera* [110].

Although the metabolomics platform enables one to gain a comprehensive view of root exudate compositions, an amalgamation of genotypic data derived from genome-wide association studies (GWAS) paves the way to dissect the genetic architecture responsible for qualitative and quantitative variations in metabolic phenotype [111]. Over the past years, owing to advances in genotyping and sequencing technology, the GWAS approach has evolved into an integrated platform for mapping and identifying the candidate gene loci responsible for natural variation in agronomically important traits, such as yield and grain quality traits in winter wheat genotypes [112], salinity tolerance in soybean [113], resistance against rust in maize [114], and drought, salt and disease resistance in alfalfa [115]. A GWAS study in *Arabidopsis* demonstrated that host-genetic factors involved in cell wall formation and synthesis, defense, and kinase activity control their phyllosphere microbiome composition [116]. These big-picture research studies motivated the conjugation of metabolomics and GWAS (mGWAS) to dissect the genetic architecture of the metabolome influencing the microbial community composition in important crop species, including rice [117], wheat [45], and maize [118].

It is increasingly recognized that the structure, dynamics, function, and response of the plant microbiome are intimately intertwined with the PSMs. Application of mGWAS for high-confidence gene identification associated with PSAMs could identify selectable traits to enhance agricultural productivity. Thus, mGWAS could resolve the following long-standing questions related to specific variations in root exudation among plant species and genotypes and correspondingly microbial composition: (1) What proportion of measurable concentrations of metabolites are controlled by genetic loci, (2) how many candidate gene loci of PSMs and how much of the genetic variance of these loci are involved in determining microbial composition, (3) to what extent are PSMs correlated with significant associations with microbiome composition, and (4) how has domestication resulted in anatomical or metabolic trade-offs that plants uniquely utilize for determining the composition of their microbiome?

## 7. Metabolomics: The Bridge of Multi-Omics in Metabolite Discovery in Plant–Microbe Interactions

The post-genomics era has seen profound progress in cutting-edge technologies for high-throughput DNA sequencing (genomics), gene expression analysis (transcriptomics), and protein analysis (proteomics). Nevertheless, the polymeric nature of proteomics and transcriptomics methods has facilitated the development of metabolomics for comprehensive, unbiased, high-throughput analyses of plant metabolome biosynthetic products, which significantly impacts plant growth and development, stress response, and ecological interactions of plants with weeds and other micro- and macro-organisms [119]. In this context, metabolomics techniques extend far beyond large-scale metabolite profiling and enable the creation of a metabolomic network atlas of the plant using mutant or transgenic lines, gene functions or impact in the metabolic pathway, thus providing a sophisticated platform to gain unique insights into metabolite changes in response to environmental and genetic modification as well as the role of metabolites in biodiversity and plant–microbe interactions [120], which is usually challenging to accomplish using traditional assays such as transcriptomics and microarray [121]. Detection and quantification of diverse classes of plant metabolites using less than three orthogonal methods is precluded owing to their varied chemical and physical properties. In this context, metabolomics analysis should be properly designed to encompass the detection of a wide range of compounds for comparative analysis and a more comprehensive understanding of exudate composition and its role in plant–microbe interaction.

### 7.1. Instrumentation in Metabolomics

Numerous tools and techniques are available for metabolite identification, screening, and quantification, each with its advantages and disadvantages. Multiple technologies can be used in tandem to profile an entire extract, and overlapping data should be combined to create a complete profile [122]. The type of extraction method used, such as liquid–liquid extraction or solid-phase extraction, etc., can affect recovery, reproducibility, and specificity in metabolite detection, thus necessitating the development of optimal extraction methods [123]. The selection of extraction solvents is also crucial. For instance, carbohydrates can be easily extracted in methanol–water, whereas lipids are best recovered in chloroform. Lastly, samples are desiccated in a sample concentrator, then reconstituted in the appropriate solvent, and if required, derivatized and injected into the instrument [124,125].

Spectrophotometric methods are considered rapid and cost-effective methods for fingerprint analysis of samples. Spectrophotometers can detect metabolites on their respective wavelength. Fourier transform infrared (FTIR) spectroscopy is another fingerprinting technique that can analyze thousands of samples of a wide range of metabolites per day without damaging or wasting the samples. In FTIR spectroscopy, metabolites absorb IR radiation and vibrate in a particular way that is specific to that metabolite. Different classes of metabolites can be detected in a specific IR range, for example, polysaccharides in 1000–1150 cm^−1^, mixed region 1250–1450 cm^−1^, amides 1600–1800 cm^−1^, and fatty acids 2800–3050 cm^−1^ [126]. Over recent decades, FTIR spectroscopy has been increasingly used in the measurement of compositional and structural changes in soil bacteria in response to plant signals [127], identification and classification of microorganisms [128], differentiation of roots of different species for a deeper understanding of the belowground root interaction [129,130], chemical changes induced in roots after microbial colonization [131], and the influence of root distribution patterns on other plant species [132].

Nuclear magnetic resonance (NMR) spectroscopy has emerged as a powerful analytical technique for metabolomics and structural studies due to minimal sample preparation, preservation of biological integrity of samples during analysis, simultaneous quantification of metabolites in a complex mixture, and high experimental reproducibility [133]. In NMR, electrically charged nuclei in a strong constant field are perturbed by an external magnetic field. The resulting interaction of the magnetic moment of an atomic nucleus produces the phenomenon of magnetic resonance, which can be used either to match against spectral libraries or to directly infer the basic structure to identify metabolites of biological origin [134]. Novel methods for signal amplification and artifact suppression in NMR spectra have improved the identification and quantification of primary and secondary plant metabolites [135]. Moreover, NMR-based metabolomics proved to be a promising tool for characterizing plant ecotypes based on metabolite fingerprinting [136], for identifying metabolites involved in host plant resistance to pathogens [137], to elucidate the root metabolome response of citrus and orange plant varieties against *Candidatus* Liberibacter asiaticus infection [138], and to metabolically cross-talk during plant–microbe interactions in the rhizosphere [139].

Mass spectrometry, in conjunction with chromatographic approaches, is commonly used for metabolic profiling, particularly in plant metabolomics. Liquid chromatography–tandem mass spectrometry (LC-MS/MS) methods are developed by infusing the metabolites standards into the MS, and ions are generated and extracted into the MS analyzer region, where they are detected and separated according to their mass-to-charge (m/z) ratio and are reported as mass spectra. In quantitative analysis, electrospray (ESI) or atmospheric pressure chemical ionization (APCI) techniques are commonly employed as ionization sources. ESI is a soft ionization technique employed in either positive (ES+) or negative ion mode (ES−) based on the nature of the analyte. Electrospray ionization data are transformed to molecular weight determination. Different types of mass analyzers such as quadrupole, time of flight (TOF), magnetic sector, electrostatic sector, quadrupole ion trap, and ion cyclotron resonance are used in metabolite identification [140,141]. In tandem-MS (MS/MS) metabolites are fragmented by collision with an inert gas such as argon or nitrogen, resulting in collision-induced dissociation. Tandem MS approaches typically employ a single quadrupole mass filter connected in tandem with a TOF detector (Q-TOF) [142]. This method was used for isoflavone profiling in kudzu root [143], rhizobial cytokinin production in legume roots [144], metabolic alterations induced by bioeffectors in tomato roots [145], and alkaloidal metabolome and the composition of root microbiota in *Aconitum vilmorinianum* [146]. However, triple quadrupole MS/MS technology provides even greater ion selectivity. Furthermore, many tandem MS analyses could be performed sequentially using an ion trap MS to select and capture appropriate ions. This MS approach could lead to enhanced sensitivity in metabolite identification, root microbiome characterization, and improved structural analysis [147].

To enhance chromatographic performance in terms of efficiency, resolution, robustness, and sensitivity, ultra-pressure liquid chromatography (UPLC) was developed as an alternative to high-pressure liquid chromatography (HPLC) [148]. The column packing, diameter, and particle size in the column directly affect the resolving power of the LC-MS instruments. The sensitivity of the instrument is controlled by the column, MS technique, and the analyzer used [149]. The UPLC-MS technique has been shown to be a platform for hyphenated microseparation for metabolomic analysis of teas [150] and offers unique advantages in the extraction and purification of the diffusible signal factor family used as quorum-sensing signals in rhizosphere colonization [151].

Gas chromatography–mass spectrometry (GC-MS) combines the features of gas chromatography and mass spectrometry. It provides a good balance of sensitivity and reliability for large-scale, nontargeted metabolite profiling compared to NMR and liquid chromatography–mass spectrometry [152]. Electron impact (EI) ionization is used in GCMS analysis, where the electron impact results in the removal of an electron from each metabolite, resulting in a 1^+^ ion. During this ionization process, the liquid part of the sample is first evaporated, followed by the movement of particles into the ionization source, where it is bombarded with electrons, resulting in the ionization of molecules [153]. Chromatographic separation in GC is achieved by temperature fluctuations inside the column oven using a gradient method [153]. Specialized software (MetAlign) is utilized for automated baseline correction and alignment of all extracted mass peaks across all samples, yielding detailed information about the relative abundance of hundreds of metabolites [154]. Mass spectra can be easily compared with available online libraries [155] and databases such as the National Institute of Advanced Industrial Science and Technology (AIST) spectral library and the National Institute of Standards and Technology (NIST) library or, more commonly, via in-house libraries [156,157]. The GC-MS has been used to track changes in the fraction of polar metabolites of sea bream [158] and identify phytoconstituents present in the root exudates of *Corbichonia decumbens* [159] and chemometric profiles of root extracts of *Rhodiola imbricata* [160]. Increased spatial resolution of sampling combined with GC-MS-based analysis characterized four fungal endophytes from the genus *Aspergillus* and seven bacterial endophytes from the *Kocuria*, *Bacillus*, *Arthrobacter*, *Staphylococcus*, and *Micrococcus* genera [161].

Another innovative technique is LC-NMR-MS, which can detect, quantify, and separate metabolites and combines high-throughput NMR screening with the high sensitivity of LC-MS for metabolite detection [162]. Capillary electrophoresis mass spectrometry (CE-MS) is excellent for the separation, screening, identification and quantification of numerous polar metabolites in both positive and negative ionization modes [163]. Problems related to mass resolution and accuracy can be solved by Fourier transform ion cyclotron resonance mass spectrometers (FT-ICR-MS), which can readily resolve metabolite peaks, except for stereoisomers of identical mass, which require prior chromatographic separation [164]. The Orbitrap Fourier Transform Mass Spectrometer (OrbiTrap-FT-MS) is a recent development that provides more robust and rapid sample analysis at low cost with good resolution. In contrast, MALDI-MS imaging and direct analysis in real-time mass spectrometry (DART-MS) are more user-friendly techniques with rapid pre-screening capabilities [165].

### 7.2. Instrumental Data Conversion to Meaningful Results

Before interpreting the results of metabolomic studies, the data generated should pass through four critical steps: (i) statistical processing and comparison of the raw data sets, (ii) selection of the most critical metabolite variables by data mining or knowledge discovery in data (KDD), (iii) user-friendly presentation/storage of the data, and (iv) construction of a database [166].

In raw data processing, instruments can give erroneous results when chromatographic peaks leak, peaks shift, and retention times change due to the formation of adducts, etc., and they can alter the mass-to-charge ratio of metabolites. It is vital to validate the method used for sample analysis to account for instrumental errors and the parameters of accuracy, precision, linearity, stability, and recovery. Different data sets can also be compared if they are from other MS runs or from different instruments using a range of software packages for metabolomics, spectral correction and conversion, chromatogram alignment [167], baseline correction, noise reduction, chromatogram alignment [168], or sample alignment algorithms for deconvolution [169].

Data mining uses statistical techniques and computer-based statistical applications such as SPSS (Statistical Package for Social Sciences). A GC-MS profile typically contains up to 500 variables; an FTIR profile contains thousands of variables, and t tests, chi-square distributions, or f tests, etc., are typically used to analyze the data. Furthermore, principal component analysis (PCA) and independent component analysis (ICA) are rapid analysis tools for summarizing and comparing large data sets without compromising with minimal information loss. Discriminant function analysis is typically used to evaluate plant data with high variability (DFA). Metabolomics data can also be assessed using genetic algorithm (GA) and genetic programming (GP) analyses, which are based on evolutionary genetics and Darwinian natural selection, respectively [170].

## 8. Future Perspectives

In conclusion, there is a wealth of information on allelopathic interactions between crops and weeds and some initial studies on the role of soil microbes in these interactions. However, studies integrating in-depth metagenomics analyses in the context of crop–weed interactions are lacking to date. There is a substantial need to discern the role of the microbiome in influencing plant–plant interactions in order to comprehend the interdependencies in this complex relationship that shapes both natural and cultivated plant communities for better agricultural productivity. In recent years, there has been a surge of interest in exploring the contribution of endophytic microbes to eco-friendly and cost-effective bioremediation, phytoremediation and rhizoremediation. Numerous studies have demonstrated that the colonization of plants by endophytes is beneficial because it promotes plant growth and plays a significant role in bioremediation. Endophytes were described to be enriched with those bacterial species or genotypes harboring more catabolic genes compared to the rhizosphere. The enrichment of endophytes depends on plant species and the contaminant habitats of the plant. Since allelochemical production is genetically regulated and their concentration varies between cultivars of the same plant species, it is still enigmatic how plants defend themselves from the auto-toxicity of allelochemicals they produce. Therefore, it would be intriguing to explore the involvement of endophytes in allelochemical detoxification to shield plants from the auto-toxicity of allelochemicals and whether the flexible and specific nature of selective pressure plants operate to control the composition and enrichment of the endophytic microbial community among different cultivars, which vary considerably in allelochemicals production as a dynamic and responsive mechanism to protect themselves from auto-toxicity of allelochemicals.

In summary, we expect that the integration of systems biology with complex omics data sets will enable us to tap deeper into the belowground chemical communication of plants with microorganisms and shed more light on the black box soil microbial diversity and rhizosphere interactions. In addition, with enhanced research and training in cutting-edge omics technologies, it will be tempting to speculate that the involvement of large-scale genome editing of targeted alleles of biosynthetic pathways and leveraging the breeding programs to incorporate the entire biosynthetic pathway could be used for rational manipulation of root chemistry to produce targeted secondary metabolites that recruit a beneficial microbial strain and deter pathogens. Actionable insights from such explorations can improve plant fitness and increase crop productivity.

## Figures and Tables

**Figure 1 cells-11-03254-f001:**
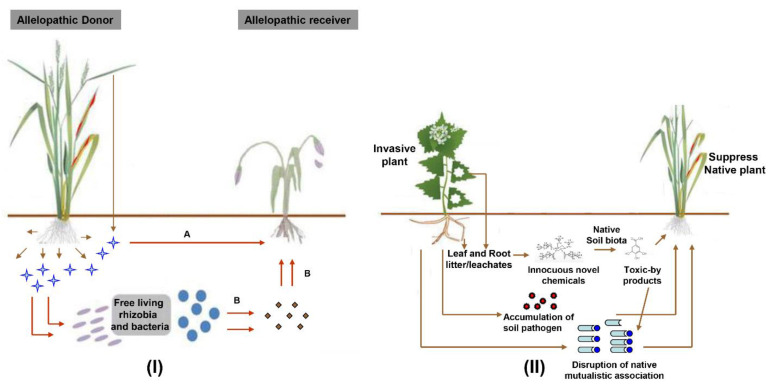
The concept of allelopathy and plant–soil interaction. (**I**) Donor plants release allelochemicals from their roots or decomposing leaf materials. Allelochemicals exert growth inhibitory activity on target plants either directly (A) or via the intermediate of soil microorganisms, which convert the plant-derived compound into a more active form (B); (**II**) Invasive plants might alter native plant–soil interactions by changing soil chemistry, the composition of soil-living symbiotic mutualists, or the abundance of local pathogens.

**Figure 2 cells-11-03254-f002:**
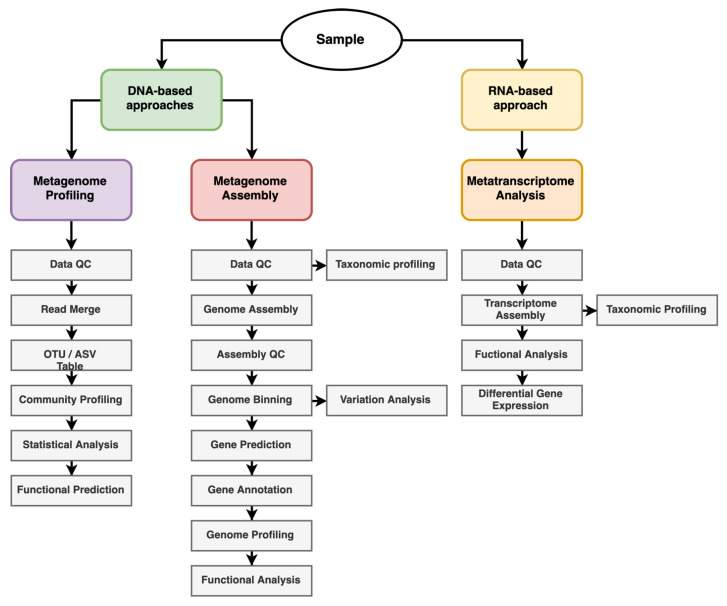
Overview of metagenome and metatranscriptome analysis workflow.

## Data Availability

Not applicable.

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
