# Peer review of "Tapping into Plant–Microbiome Interactions through the Lens of Multi-Omics Techniques"

_cells, 2022, doi:10.3390/cells11203254_

Round 1

Reviewer 1 Report

This paper summarize the researches about plant-microbiome interactions in the background of development of multi-omics techniques

Author Response

Reviewer 1 Comments

Comments and Suggestions for Authors

This paper summarizes the researches about plant-microbiome interactions in the background of rapid progress of multi-omics techniques. The contents of paper are generally comprehensive and may be helpful for those researchers studying plant microbiome interactions. However, there are still some points that need clarifying.

Response: We would like to convey our immense thanks to the reviewer for sparing their valuable time to enhance the quality of the manuscript. We have taken into consideration of all minor concerns and incorporated the suggested changes in the revised review. 

  1. The title of this paper is “Tapping into plant-microbiome interactions through the lens of multi-omics techniques”. So I think authors want to emphasize the researches about new progresses using omics techniques. However, Part1, 2 and 3 that introduced some basic concepts occupied great length, which should be shorten and become more easily readable.

Response: The theme of review has been constructed based on the past and current discoveries in plant-microbes and plant-plant interaction, followed by instrumentation widely implicated in characterizing the secondary metabolite. These sections will assist readers in gaining thorough discoveries in plant microbial interaction. We believe this section forms the foundation of the review, and removing the soul of the review will disrupt the content and flow of the review and it will be difficult for new researches in the field to comprehend the real essence of this review. In order to maintain the flow of review, we would like to request the reviewer to allow us to retain these sections.

  1. The paper mainly introduced researches about fungi and bacteria. Are there some progresses about virus?

Response: We appreciate the interesting fact raised by the reviewer. However, over the past years, fungus and bacteria in plant-microbes interaction have received extensive attention due to their mutualistic benefits. Plant-virus interaction has been discussed extensively in the context of biotic stress, which is not the focused area of this review. But the point is well noted, and we will incorporate viruses in the future review as a separate topic in the context of interaction of metabolites with parasitic species.

  1. A table summarizing the words from line 216 to 221 will benefit readers.

Response: We have enumerated the list of allelochemicals so that readers can distinguish the classes of allelochemicals

  1. There are some grammar errors in the paper. Such as:

Line 115: such as nitrogen (N), phosphorus (P), and potassium (K) sulfur (S) for

Line 141: facilitate--facilitates

Line 171: the interaction of PSM and microbes-- the interaction between PSM and

microbes

Line 176: researched—studied

Line 197: A. thaliana should be italic

Line 225:amounts—amount

Line 287:effort—efforts

Line 488:2800-3050cm-1-- 2800-3050 cm-1

Response: We apologize for the typo in the review. We have made the suggested changes. In addition, we enlisted the help of an English proofreading service. In the revised version, the major corrections have been highlighted in track change mode.

Reviewer 2 Report

In this review, the authors have highlighted the different methodologies adapted to study the symbiotic interaction of plant roots with the microbiota. The authors have discussed how the advent of next-gen sequencing methods has allowed researchers to study the root composition and interactions of the microbiota with its plant host. The primary and secondary metabolites can be used to design targets to treat disease-spreading microbiota. The authors have also discussed the use of metabolomics to identify novel molecules in root exudates. Overall, the review is well written and should of importance to people studying plant-microbiota interactions.

Author Response

Reviewer 2 Comments

Comments and Suggestions for Authors

In this review, the authors have highlighted the different methodologies adapted to study the symbiotic interaction of plant roots with the microbiota. The authors have discussed how the advent of next-gen sequencing methods has allowed researchers to study the root composition and interactions of the microbiota with its plant host. The primary and secondary metabolites can be used to design targets to treat disease-spreading microbiota. The authors have also discussed the use of metabolomics to identify novel molecules in root exudates. Overall, the review is well written and should of importance to people studying plant-microbiota interactions.

Response: Thank you very much for your valuable feedback and for accepting the review. We are highly grateful for the endorsement of the review article for publication

Reviewer 3 Report

Overall the present study is well executed yet, the minor issues need to resolve before publication.

1. Abstract keywords are too lengthy try to write any catchy single word keywords which should reflects the theme of your research.

2. Page 3 line 109 Aim of the research needs to rewritten addresing the importance and novelty of the present review and which way it is unique  from the past research.

3. Page10 line429 title is too lenghty needs to rewrite.

4. Page 14 line 629 future perspectives of the study needs to be rewritten. 

5. Overall the complete paper needs to be grammar check and typo errors needs to be corrected.

Author Response

Reviewer 3 Comments
Comments and Suggestions for Authors
Overall, the present study is well executed yet, the minor issues need to resolve before publication.
Response: Thank you so much for your insightful positive comments. We have considered all suggested corrections and incorporated in the revised version of the Review
1. Abstract keywords are too lengthy try to write any catchy single word keywords which should reflects the theme of your research.
Response: We have replaced the lengthy keywords with short, catchy single words.
2. Page 3-line 109 Aim of the research needs to rewritten addressing the importance and novelty of the present review and which way it is unique from the past research.
Response: We have highlighted the novelty of our review in lines 110-121
3. Page10 line429 title is too lengthy needs to rewrite.
Response: Subtitle has been shortened
4. Page 14-line 629 future perspectives of the study needs to be rewritten.
Response: The suggested line of future perspective has been rewritten in order to make sentence more transparent to readers
5. Overall the complete paper needs to be grammar check and typo errors needs to be corrected.
Response: The manuscript was proofread by an English proofreading service. The major correction we have highlighted is in “track changes mode”.
